# Critically-Ill Patients with Biliary Obstruction and Cholangitis: Bedside Fluoroscopic-Free Endoscopic Drainage versus Percutaneous Drainage

**DOI:** 10.3390/jcm11071869

**Published:** 2022-03-28

**Authors:** Yi-Jun Liao, Wan-Tzu Lin, Hsin-Ju Tsai, Chia-Chang Chen, Chun-Fang Tung, Sheng-Shun Yang, Yen-Chun Peng

**Affiliations:** 1Division of Gastroenterology and Hepatology, Department of Internal Medicine, Taichung Veterans General Hospital, Taichung 40705, Taiwan; s19001029@gmail.com (Y.-J.L.); woantyy0025@gmail.com (W.-T.L.); a9194024@hotmail.com (H.-J.T.); judgespear@gmail.com (C.-C.C.); et@vghtc.gov.tw (C.-F.T.); yansh2525@gmail.com (S.-S.Y.); 2School of Medicine, Chung Shan Medical University, Taichung 40201, Taiwan; 3Ph.D. Program in Translational Medicine, National Chung Hsing University, Taichung 40227, Taiwan; 4Institute of Biomedical Sciences, National Chung Hsing University, Taichung 40227, Taiwan; 5Department of Internal Medicine, Taichung Veterans General Hospital Chiayi Branch, Chiayi 60090, Taiwan; 6School of Medicine, National Yang Ming Chiao Tung University, Taipei 11230, Taiwan

**Keywords:** acute cholangitis, critically ill, endoscopic biliary drainage, percutaneous biliary drainage

## Abstract

Severe acute cholangitis is a life-threatening medical emergency. Endoscopic biliary drainage (EBD) or percutaneous transhepatic biliary drainage (PTBD) is usually used for biliary decompression. However, it can be risky to transport a critical patient to the radiology unit. We aimed to compare clinical outcomes between bedside, radiation-free EBD and fluoroscopic-guided PTBD in patients under critical care. Methods: A retrospective study was conducted on critically ill patients admitted to the intensive care unit with biliary obstruction and cholangitis from January 2011 to April 2020. Results: A total of 16 patients receiving EBD and 31 patients receiving PTBD due to severe acute cholangitis were analyzed. In the EBD group, biliary drainage was successfully conducted in 15 (93.8%) patients. Only one patient (6.25%) encountered post-procedure pancreatitis. The 30-day mortality rate was no difference between the 2 groups (32.72% vs. 31.25%, *p* = 0.96). Based on multivariate analysis, independent prognostic factors for the 30-day mortality were a medical history of malignancy other than pancreatobiliary origin (HR: 5.27, 95% confidence interval [CI]: 1.01–27.57) and emergent dialysis (HR: 7.30, 95% CI: 2.20–24.24). Conclusions: Bedside EBD is safe and as effective as percutaneous drainage in critically ill patients. It provides lower risks in patient transportation but does require experienced endoscopists to perform the procedure.

## 1. Introduction

Severe acute cholangitis is a life-threatening medical emergency. To save the lives of such critically ill patients, early biliary drainage is a necessary procedure [1,2,3]. Both endoscopic biliary drainage (EBD) and percutaneous transhepatic biliary drainage (PTBD) reduce mortality from acute cholangitis [4,5]. In most hospitals, fluoroscopy is not available in the intensive care unit (ICU). Transport of those hemodynamically unstable patients who are connected with a mechanical ventilator to the fluoroscopic room is an extremely risky process. However, performing bedside EBD without fluoroscopic guidance is known to be feasible [6,7,8,9,10].

The Tokyo guidelines recommend that endoscopic transpapillary biliary drainage be the first-line procedure for severe acute cholangitis. Despite the risk of post-endoscopic retrograde cholangiopancreatography (ERCP) pancreatitis, EBD has fewer adverse events than the percutaneous drainage procedure [4]. PTBD is positioned as an alternative procedure when EBD is infeasible or has previously failed. For critically ill patients, the efficacy and safety of emergent, radiation-free EBD has been demonstrated [9,11]. However, it still remains a challenge for endoscopists to confirm accurate biliary cannulation without fluoroscopic guidance in the ICU. Several techniques, such as portable X-ray, trans-abdominal ultrasound and intraductal ultrasound have been reported as helpful in radiation-free ERCP for confirming properly performed biliary cannulation [7,8,10,12].

To date, no study has compared the various biliary drainage approaches for critically ill patients. Here, we aimed to compare the clinical outcomes of bedside radiation-free EBD with conventional PTBD for critically ill patients with acute, severe cholangitis.

## 2. Materials and Methods

### 2.1. Study Design

We conducted a retrospective study by enrolling patients who had been admitted to the intensive care unit (ICU) in Taichung Veteran General Hospital with severe biliary tract infections and respiratory failure during an approximately 10-year period from January 2011 to April 2020. We first reviewed their medical charts, laboratory data, image finding, radiological interventions and endoscopic findings. Our study protocol was approved by the Institutional Review Board of Taichung Veterans General Hospital (No: CE21470A).

### 2.2. Study Subjects

Diagnosis of biliary tract infection was made according to clinical manifestations, laboratory data regarding cholestasis and image findings of biliary tract dilation. Severe biliary tract infection was graded according to the Tokyo guidelines 2018 and defined as being acute cholangitis associated with dysfunction in at least one organ [13]. Respiratory failure was defined as a patient having hypoxemia, and endotracheally intubated with ventilator support. In our hospital, the percutaneous trans-hepatic biliary drainage (PTBD) was typically used for biliary decompression. Some patients received bedside EBD in the ICU without fluoroscopy due to various contraindications for PTBD insertion. The indication and method of biliary drainage were decisions made by physicians, based upon their clinical judgment. All patients signed off on informed consents for the invasive drainage procedures.

### 2.3. Pre-Drainage Preparation

The Acute Physiology and Chronic Health Evaluation II (APACHE II) score was calculated within 24 h after ICU admission as a routine evaluation protocol at our hospital [14]. For patients with coagulopathy characterized by a prolonged prothrombin time of an international normalized ratio (PTINR) ≥ 1.5, blood transfusion was given with fresh frozen plasma. In the event of low platelet counts (≤50 ×10^3^/μL), platelet transfusion was performed prior to the drainage procedure. The time to biliary drainage was defined as the time interval between arrival at the emergency department or the onset of cholangitis during admission and the procedure of biliary drainage. 

### 2.4. PTBD Procedures

For PTBD, critically-ill patients were transported to the fluoroscopic room under full monitors, where the gallbladder and biliary system were sonographically examined by a radiologist. After sterilization and local anesthesia, a needle was punctured into the gallbladder or intrahepatic bile duct. The needle was then removed to allow the outflow of bile. The contrast medium was then injected to confirm the puncture location. A guidewire was inserted along the needle sheath. Finally, a pigtail tube was inserted along the guidewire into either the gallbladder or biliary tract.

### 2.5. EBD Procedures

With the patient placed in the left decubitus position, bedside EBD was performed using a standard side-view duodenoscope (TJF-260; Olympus Optical Co., Ltd., Tokyo, Japan). Wire-guided biliary cannulation was performed over the major papilla using a standard catheter or sphincterotome. A syringe was used to aspirate the bile. The free flow of bile and visualization of bile in the tip of the ERCP catheter/sphincterotome were used to confirm a successful cannulation into the common bile duct. The guidewire was advanced into the bile duct as deeply as possible. Then, endoscopic Retrograde Biliary Drainage (ERBD) was inserted under the guidance of a guidewire. Portal X-ray then documented the final position of ERBD. The technical success rate was defined as success in accomplishing ERBD insertion. 

### 2.6. Outcome Measurement

After the biliary drainage procedure, parameters including complete blood count, serial liver, renal function, amylase, lipase and PTINR were measured. Adverse events of EBD, such as bleeding, perforation and post ERCP-pancreatitis were recorded. The 30-day mortality rate was evaluated.

### 2.7. Statistical Analysis

Statistical analyses were performed using the Statistical Package for the Social Sciences (version 22.0; SPSS, Inc., Chicago, IL, USA). Categorical variables presented as numbers and percentages were compared using the Chi-square or Fisher’s exact test. Continuous variables presented as median and interquartile range (IQR) were compared using the Mann–Whitney U test. Kaplan–Meier survival curves of the two drainage groups were plotted and compared with the log-rank test. Prognostic factors associated with 30-day mortality were analyzed by the Cox proportional hazard model. Two-tailed *p* values < 0.05 were considered statistically significant.

## 3. Results

### 3.1. Study Subjects 

A total of 50 patients were admitted to the ICU having the diagnosis of severe biliary tract infection with respiratory failure. Three of the patients were excluded due to an absence of biliary drainage. These three patients had extremely unstable vital signs even under aggressive resuscitation, high dose vasopressor and broad-spectrum antibiotics usage. They subsequently died before biliary drainage could be performed. Ultimately, we analyzed 16 patients receiving EBD and 31 patients receiving PTBD. The flowchart of patient recruitment is shown in Figure 1. Patients received EBD due to the following causes: five patients experienced mild IHDs dilatation and difficulty in PTBD approach, two had moderate to massive ascites, seven had thrombocytopenia or coagulopathy and two experienced profound shock under high levels of vasopressors.

### 3.2. Baseline Characteristics

As shown in Table 1, no inter-group difference was found in terms of patients’ baseline characteristics. Most patients were of an elderly age with chronic diseases. No inter-group difference was found in terms of median APACHE2 score (EBD vs. PTBD, 26.5 vs. 27, *p* = 0.71). No inter-group differences were found in terms of severe biliary tract infection complicated with septic shock, bacteremia, acute renal failure, emergent hemodialysis, thrombocytopenia, or coagulopathy. Time to biliary drainage was defined as the duration from arrival at the emergency department or the onset of cholangitis during admission to the procedure of biliary drainage. The time to biliary drainage was significantly longer in the EBD group (EBD vs. PTBD, 3.25 days vs. 1 day, *p* = 0.01). 

### 3.3. Clinical Data

The laboratory data associated with cholangitis prior to the drainage procedures were also similar between EBD and PTBD, except that alanine aminotransferase (ALT) was higher in the PTBD group (EBD vs. PTBD, 34 vs. 88, *p* = 0.03). As shown in Table 2, the white blood count, total bilirubin and change in total bilirubin (ΔTotal bilirubin) after biliary drainage were similar among the two groups.

### 3.4. ERCP Procedures

As shown in Table 3, indications of bedside ERCP were as follows: common bile duct (CBD) dilatation with cholangitis (*n* = 5, 31.25%), choledocholithiasis (*n* = 8, 50%), gallstone pancreatitis (*n* = 1, 6.25%), post-operative bile leakage (*n* = 1, 6.25%) and ampulla vater tumor (*n* = 1, 6.25%). ERCP procedures were performed on 16 patients. Cannulation and ERBD were successfully conducted in 15 (93.75%) patients, except for 1 patient with an inaccessible papilla due to severe hiatal hernia. Endoscopic sphincterotomy (EST) was performed on one patient. The median procedure time was 20 min. Only one patient (6.25%) developed mild post-ERCP pancreatitis. In the PTBD group, two patients (6.45%) encountered procedure related bleeding, which was controlled through medical treatment. 

### 3.5. Risk Factors for 30-Day Mortality Rate

As shown in Figure 2, the overall 30-day mortality rate was similar between the two groups (EBD 31.25% and PTBD 32.72%; *p* = 0.96). We found that the two independent prognostic factors of 30-day mortality were medical history of malignancy other than pancreatobiliary origin (HR: 5.27, 95% confidence interval [CI]: 1.01–27.57) and emergent dialysis (HR: 7.30, 95% CI: 2.20–24.24), as shown in Table 4. 

## 4. Discussion

Our results, although obtained by an experienced endoscopist in a small number of patients and in a single center, suggest that bedside radiation-free EBD could be as safe and effective as PTBD for biliary decompression in critically ill patients. The two approaches displayed similar 30-day mortality rates. Those patients who had a malignancy other than of pancreatobiliary origin or acute kidney injury requiring emergent dialysis experienced higher short-term mortality regardless of the drainage procedure.

Patients experiencing acute cholangitis with septic shock and multiple organ failure face risks during transportation, although emergency biliary decompression is still necessary [3]. Previous studies have reported on the benefits of tools such as abdominal sonography, portable X-ray or IDUS following cannulation when confirming the exact location of the guidewire [7,8,10,15]. The success of EBD in radiation free ERCP reaches 88–92% even without such assistive tools [9,11]. Our study showed a high technical success rate (93.75%) in the EBD group without implementing any assistive tools. Failed cannulation occurred in only one patient due to inaccessibility to the papilla. Radiation-free EBD performed by an experienced endoscopist is important for successful biliary cannulation. We confirmed CBD cannulation through bile aspiration and then advanced the catheter along with the guidewire deep into the CBD without experiencing any resistance. Most of our cases involved distal obstruction or choledocholithiasis-related cholangitis. The use of endoscopic biliary drainage tube (5 to 7 cm) allowed for successful drainage. Alternatively, for the treatment of proximal stenosis or tumor-related obstruction, ERCP with fluoroscopic guidance is mandatory in order to assure the exact location and length of the stricture [9].

EBD can be performed under various conditions when PTBD is contraindicated or difficult, such as the occurrence of ascites, bleeding tendency, and an inaccessible bile duct [4]. EBD could also preserve normal digestion and minimize the risk of transportation. The Tokyo guidelines recommend the endoscopic biliary drainage procedure as the first line of treatment, while PTCD can be an alternative procedure when selective cannulation has failed [4,16,17]. Based on our results, this recommendation can also be applied to ICU patients.

Post-ERCP complications are mainly bleeding, perforation, post-ERCP pancreatitis and cholangitis [18]. A bedside EBD study on 26 patients conducted by Wang et al. reported zero complications related to the procedure [9]. Another similar study by Hong et al. reported only 1.25% (1/80) of patients had developed mild pancreatitis [11]. In our study, we found that only one patient (6.25%) had developed mild post-ERCP pancreatitis. In summary, radiation-free ERCP is safe with no additional complications having been encountered for critically ill patients in the ICU. There were no major adverse events except for mild pancreatitis.

Endoscopic nasal biliary drainage (ENBD) may also be used for bedside biliary drainage with the benefit of monitoring drainage function. However, ENBD often encounters certain problems such as obstruction due to sludge or the presence of pus as it is related to a small internal diameter and an impaired digestive function coming as the result of bile divergence. Electrolyte imbalance and dehydration may also be induced by ENBD [19]. Additionally, ENBD may be accidentally dislodged or removed by a confused patient [20]. In cases of severe acute cholangitis, previous studies have reported no difference between ENBD and ERBD in terms of their efficacy [19,20,21,22]. Wang et al. reported that ENBD is a better form of treatment than ERBD, as the length and diameter of the plastic stent cannot be determined in the absence of the fluoroscope [9]. Further studies are still needed to verify the results of bedside ERCP seen in our current small-sample study.

Bedside ERCP has reported a 30-day mortality rate of 36.25% and that APACHE II score ≥ 22 is an independent risk factor for mortality [11]. In our EBD group, the median APACHE II score was 26.5 and the 30-day mortality rate was 31.25. Risk factors of 30-day mortality were underlying malignancies and acute renal failure requiring emergent hemodialysis other than the APACHEII score. Delayed biliary decompression >12–48 h in acute severe cholangitis patients is a risk factor for mortality [2,23]. Early biliary drainage occurring within 24 h shortens the length of ICU stay but does not improve survival [24,25]. The time to biliary drainage was, however, not a factor affecting the mortality rate in our study. Both underlying comorbidities and severity of organ failure seem to be more significant in regards to mortality. We found no statistical differences in mortality rate between the EBD and PTBD groups. This indicated that infection control with adequate biliary drainage is more critical than which drainage method is implemented [23,26].

There are several limitations to our study. First, this is a retrospective study with a small sample size. More data need to be collected in order to better consolidate our conclusions. Second, experienced endoscopists for ERCP are not always available. The practice of bedside ERCP is therefore not so easily applied to community hospitals.

## 5. Conclusions

In conclusion, bedside radiation-free EBD is a safe and effective treatment with a high success rate, while involving few complications. The 30-day mortality rate is associated with an underlying non-pancreaticobiliary malignancy and emergent hemodialysis due to sepsis-related acute renal failure. Bedside EBD could be optioned as the first choice for biliary drainage in patients with critical illness.

## Figures and Tables

**Figure 1 jcm-11-01869-f001:**
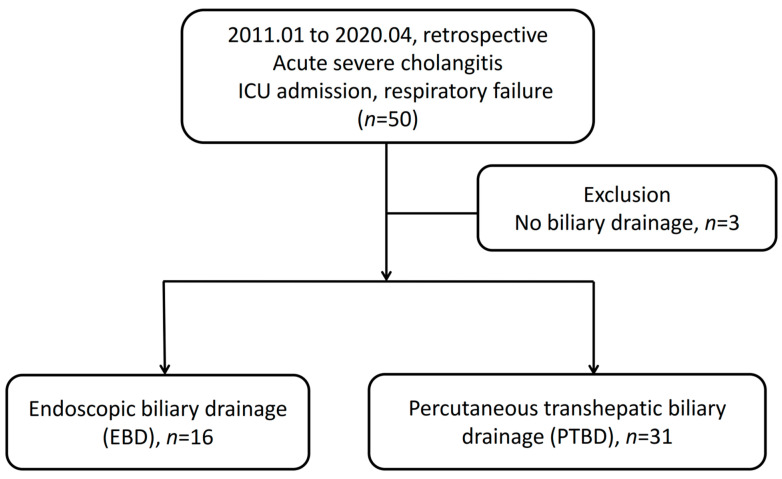
Flowchart of patients enrolled in this study. ICU, intensive care unit.

**Figure 2 jcm-11-01869-f002:**
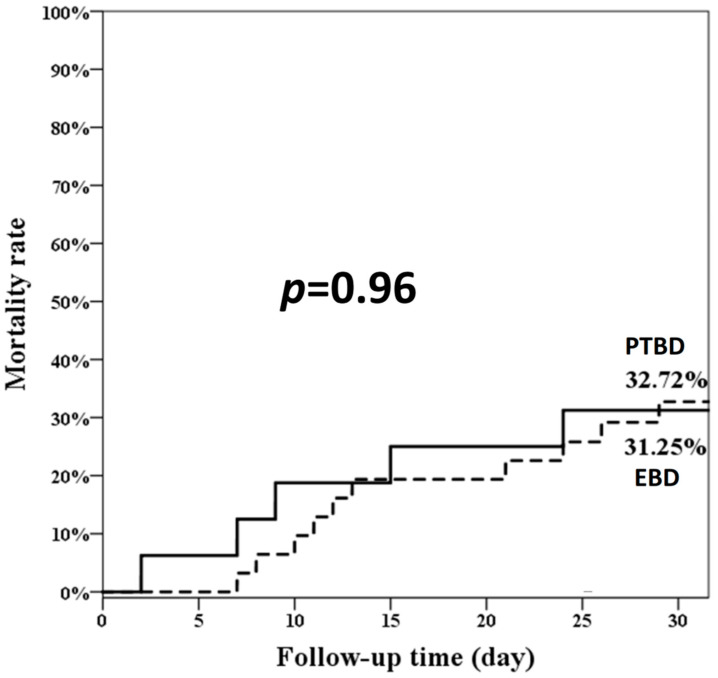
The overall 30-day mortality rate of the two groups. EBD: endoscopic biliary drainage; PTBD: percutaneous transhepatic biliary drainage.

**Table 1 jcm-11-01869-t001:** Baseline characteristics of study subjects.

Characteristics	EBD (*n* = 16)	PTBD (*n* = 31)	*p* Value
Age, median (range), years	81.60 (68.93–86.12)	78.22 (68.76–84.76)	0.74
Gender, Male (%)	9 (50.00)	15 (48.39)	1.00
APACHE II score, median (range)	28 (24.5–34.25)	27 (21–33)	0.47
Underling comorbidities, *n* (%)			
Hypertension	6 (37.50)	19 (61.29)	0.22
Type 2 diabetes mellitus	5 (31.25)	12 (38.71)	0.85
Chronic kidney disease	7 (43.75)	11 (35.48)	0.81
Congestive heart failure	6 (37.50)	5 (16.13)	0.15
Liver cirrhosis	2 (12.50)	4 (12.90)	1.00
Cerebral vascular accident	3 (18.75)	3 (9.68)	0.40
COPD	4 (25.00)	4 (12.90)	0.42
Pancreaticobiliary malignancy	1 (6.25)	4 (12.90)	0.65
Other malignancy	5 (31.25)	3 (9.68)	0.10
APACHE II score, median (range)	26.5 (23.5–33.75)	27 (21–33)	0.71
Septic shock, *n* (%)	12 (75.00)	24 (77.42)	1.00
Bacteremia, *n* (%)	5 (31.25)	16 (51.61)	0.31
Acute renal failure, *n* (%)	8 (50.00)	20 (64.52)	0.52
Emergent hemodialysis, *n* (%)	5 (31.25)	12 (38.71)	0.85
PTINR > 1.5, *n* (%)	5 (31.25)	5 (16.13)	0.41
Platelet < 100,000/mm^3^, *n* (%)	9 (56.25)	14 (45.16)	0.68
Time to biliary drainage (day)	3.5 (2–6.75)	1 (1–3)	0.01 *

* *p* < 0.05. Continuous data are presented as median (IQR, 25th–75th percentile). Categorical data are presented as numbers and percentages. EBD: endoscopic biliary drainage; PTBD: percutaneous transhepatic biliary drainage; COPD, chronic obstructive pulmonary disease; APACHE II, Acute Physiology and Chronic Health Evaluation II; PTINR, prothrombin time of an international normalized ratio.

**Table 2 jcm-11-01869-t002:** Clinical data before and after biliary drainage.

	EBD (*n* = 16)	PTBD (*n* = 31)	*p* Value
Pre-drainage			
WBC (×1000/μL)	11.73 (7.51–17.53)	16.85 (9.34–21)	0.08
Hemoglobin (g/dL)	9.05 (8.05–11.03)	10.5 (9.4–12.1)	0.03
Platelet (×1000/μL)	90 (43.25–154.50)	117 (67–196)	0.21
Total bilirubin (mg/dL)	5.45 (3.00–11.25)	4.8 (2.4–8.3)	0.45
Alk-P (U/L)	234 (121.75–416.50)	317 (166–506)	0.24
AST (U/L)	94 (57–188.5)	113 (65–239)	0.44
ALT(U/L)	36 (18.5–97.5)	88 (41–153)	0.03 *
PTINR	1.25 (1.15–1.65)	1.26 (1.17–1.36)	0.77
Creatinine (mg/dL)	2.02 (0.81–3.59)	1.69 (1.12–3.04)	0.74
Post-drainage			
WBC (×1000/μL)	10.05 (9.28–12.7)	15.73 (11.2–21.29)	0.05
Hemoglobin(g/dL)	9.35 (7.95–10.23)	9.8 (8.9–10.9)	0.20
Platelet (×1000/μL)	79.5 (47–143.25)	74 (49–124)	0.88
Total bilirubin (mg/dL)	5 (2.3–14.2)	4.1 (1.2–6.6)	0.31
ΔTotal bilirubin	0.2 (−1–2.73)	0.7 (−2–2.2)	0.75
Alk-P (U/L)	209 (143–540.75)	207 (131–382)	0.73
AST (U/L)	66 (41–184)	98.5 (40.75–194)	0.71
ALT (U/L)	20 (15–63)	74 (33–148.75)	0.02 *
PTINR	1.57 (1.16–1.81)	1.15 (1.05–1.38)	0.02 *
Creatinine (mg/dL)	1.42 (90.85–2.93)	2.06 (0.86–2.70)	0.65

* *p* < 0.05. Continuous data are presented as median (IQR, 25th–75th percentile). EBD: endoscopic biliary drainage; PTBD: percutaneous transhepatic biliary drainage; WBC, white blood count; AlK-P, alkaline phosphatase; AST, aspartate aminotransferase; ALT, alanine aminotransferase; PTINR, prothrombin time international normalized ratio; ΔTotal bilirubin, pre-drainage total bilirubin minus post drainage total bilirubin.

**Table 3 jcm-11-01869-t003:** ERCP indications, procedures and outcome in 16 patients.

	EBD (*n* = 16)
Indications, *n* (%)	
CBD dilatation, cholangitis	5 (31.25)
Choledocholithiasis	8 (50.00)
Biliary pancreatitis	1 (6.25)
Post-operation biliary leakage	1 (6.25)
Ampulla vater tumor	1 (6.25)
Procedure time (min), median (IQR)	20 (20.35)
Interventions, *n* (%)	
EPT	1 (6.25)
ERBD	15 (93.75)
Technical success, *n* (%)	15 (93.75)
Adverse events, *n* (%)	1 (6.25)
Overall 30-day mortality rate	5 (31.25)

EBD: endoscopic biliary drainage; EPT, endoscopic papillotomy; ERBD, endoscopic Retrograde Biliary Drainage.

**Table 4 jcm-11-01869-t004:** Predictive factors associated with 30-day mortality.

Variable	Univariate Analysis			Multivariate Analysis		
	HR	95% CI	*p* Value	HR	95% CI	*p* Value
Group (PTBD vs. EBD)	0.97	(0.33–2.84)	0.956			
Age	1.05	(0.99–1.12)	0.082			
Male gender	0.99	(0.36–2.73)	0.982			
APACHE II score	1.05	(0.97–1.14)	0.195			
Underlying comorbidities						
Chronic kidney disease	1.44	(0.52–3.98)	0.480			
Congestive heart failure	1.84	(0.63–5.40)	0.264			
Liver cirrhosis	2.28	(0.64–8.13)	0.202			
COPD	0.78	(0.18–3.47)	0.747			
Pancreaticobiliary malignancy	0.56	(0.07–4.28)	0.578			
Other malignancy	3.73	(1.27–11.01)	0.017 *	5.27	(1.01–27.57)	0.049 *
Septic shock	0.54	(0.18–1.58)	0.262			
Time to drainage >2 days	2.68	(0.95–7.57)	0.062			
Emergent dialysis	4.86	(1.65–14.29)	0.004 *	7.30	(2.20–24.24)	0.001 **
Pre-drainage data						
WBC (×1000/μL)	1.00	(1.00–1.00)	0.339			
Hemoglobin (g/dL)	0.92	(0.74–1.15)	0.451			
Platelet (×1000/μL)	1.00	(0.99–1.01)	0.716			
Total bilirubin (mg/dL)	0.27	(0.97–1.13)	0.273			
AlK-P (U/L)	1.00	(1.00–1.00)	0.752			
AST (U/L)	1.00	(1.00–1.00)	0.173			
ALT (U/L)	1.00	(1.00–1.00)	0.902			
PTINR	4.14	(1.18–14.57)	0.027 *	3.42	(0.68–17.32)	0.137
ΔTotal bilirubin	0.82	(0.73–0.93)	0.001 **	0.90	(0.76–1.06)	0.197

* *p* < 0.05, ** *p* < 0.01. PTBD, percutaneous transhepatic biliary drainage; EBD, endoscopic biliary drainage; COPD, chronic obstructive pulmonary disease; WBC, white blood count; AlK-P, alkaline phosphatase; AST, aspartate aminotransferase; ALT, alanine aminotransferase; PTINR, prothrombin time international normalized ratio; ΔTotal bilirubin, pre-drainage total bilirubin minus post-drainage total bilirubin; HR, hazard ratio; CI, confidence interval.

## Data Availability

Data are available on request from the authors.

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
