# Peer review of "Critically-Ill Patients with Biliary Obstruction and Cholangitis: Bedside Fluoroscopic-Free Endoscopic Drainage versus Percutaneous Drainage"

_jcm, 2022, doi:10.3390/jcm11071869_

Round 1

Reviewer 1 Report

This is a retrospective study conducted on 50 critically sick patients admitted to the intensive care unit with biliary obstruction and severe acute cholangitis.

A total of 16 patients receiving Endoscopic Biliary Drainage(EBD) and 31 patients receiving Percutaneous Transhepatic Biliary Drainage (PTBD) were analyzed.

The Authors conclude that bedside radiation-free EBD is as safe and as effective as conventional PTBD for biliary decompression in critically- ill patients. The two approaches had similar 30-days mortality rates.

Comments:

Three patients were excluded due to absence of biliary drainage. What does it mean?

The paper has many limitations as partly indicated by the authors themselves. First, this is a retrospective study with a small sample size. More data should be collected to consolidate their conclusions. Second, it was not randomized and, although the clinical characteristics of the two groups were similar before and after drainage, it cannot be excluded that the patients assigned to the EBD group were chosen because they were in better general condition. Third, experienced endoscopists for ERCP are not always available.

The discussion is far too long and verbose and should be very shortened.

The first sentence of the discussion: "Our results indicated that bedside radiation-free EBD is as safe and as effective as conventional PTBD for biliary decompression in critically- ill patients. "should be modified as follows:" Our results, although obtained by an experienced endoscopist, in a small number of patients and in a single center, suggest that bedside radiation- free EBD could be as safe and as effective as PTBD for biliary decompression in critically-ill patients. "

Reviewer 2 Report

Some points must be adressed: 

1) concerning discussion of demonstrating biliary cannulation success in non-fluoro ERCP (compare page 2, first para) aspiration of bile and visualization of bile in the tip of the ERCP catheter/papillotome should be discussed

2) study cohort and indication for ERCP

uncomplicated choledocholithiasis (and every non-cholangitis entity) is hardly an indication for ERCP under such circumstances and must be very critically discussed

3) the paper needs professional language editing
